# Therapeutic Potential of DPHC, A Brown Seaweed Polyphenol, Against TNF-α-Induced Inflammatory Muscle Loss

**DOI:** 10.3390/md23100376

**Published:** 2025-09-26

**Authors:** Minji Kim, Won-Woo Lee, Kil-Nam Kim, Young-Mog Kim, You-Jin Jeon, Fengqi Yang, Seo-Young Kim, Hyo-Geun Lee

**Affiliations:** 1Research Center for Marine Integrated Bionics Technology, Pukyong National University, Busan 48513, Republic of Korea; minjikim@pknu.ac.kr (M.K.); ymkim@pknu.ac.kr (Y.-M.K.); 2Division of Practical Research, Honam National Institute of Biological Resources, 99, Gohadoan-gil, Mokpo-si 58762, Republic of Korea; 21cow@hnibr.re.kr; 3Gwangju Center, Korea Basic Science Institute, Gwangju 61751, Republic of Korea; knkim@kbsi.re.kr; 4Department of Marine Life Science, Jeju National University, Jeju 63243, Republic of Korea; youjinj@jejunu.ac.kr (Y.-J.J.); yfq426@jejunu.ac.kr (F.Y.); 5National Marine Biodiversity Institute of Korea, 75, Jangsan-ro 101-gil, Janghang-eup, Seocheon 33362, Republic of Korea

**Keywords:** *Ishige okamurae*, pro-inflammatory cytokine, inflammatory muscle loss, swimming performance, algal polyphenol

## Abstract

Inflammatory muscle loss results from excessive inflammatory responses, causing muscle damage and weakness. In the current investigation, we evaluated the protective effects of diphlorethohydroxycarmalol (DPHC) against tumor necrosis factor-alpha (TNF-α)-induced skeletal muscle inflammation and muscle loss and elucidated the underlying mechanisms. Furthermore, the effect of DPHC on swimming performance was confirmed under TNF-α-induced inflammatory muscle loss-conditioned zebrafish by assessing the swimming number, distance moved, time spent swimming, frequency of swimming zebrafishes in an upstream swim track (Zone A). In vivo behavioral endurance test results indicated that TNF-α treatment significantly decreased the number of swimming zebrafish and swimming distance in Zone A compared with the Control. Meanwhile, the DPHC treatment significantly increased the number of swimming zebrafish and swimming distance in Zone A compared to TNF-α-induced zebrafish. These findings indicate that DPHC treatment effectively improved the swimming performance of TNF-α-induced zebrafish. In an additional study, TNF-α significantly induced inflammatory muscle loss by upregulating nuclear factor kappa light chain enhancer of activated B cells (NF-κB) mitogen activated protein kinase (MAPK) associated proteins and MuRF-1 in the skeletal muscle tissues of TNF-α-induced zebrafish. However, DPHC administration significantly counteracted TNF-α-induced inflammation and muscle loss by downregulating NF-Κb and MAPK-associated proteins, as well as the muscle degradation-related proteins MuRF-1 and MAFbx, in the skeletal muscle tissues of TNF-α-induced zebrafish. In summary, our research findings demonstrated that DPHC from *Ishige okamurae* could be used for the development of nutraceuticals or functional foods targeting inflammatory muscle loss.

## 1. Introduction

Skeletal muscle, which constitutes approximately 40% of the body’s total mass in humans, plays a crucial role in physical movement, and muscle atrophy, characterized by the loss of muscle mass, commonly occurs due to factors such as injury, nerve damage, inflammation, or aging [1]. The major factors contributing to muscle atrophy include musculoskeletal disuse following muscle injury, which is closely associated with the protein expression of muscle RING-finger protein-1 (MuRF-1), and muscle atrophy F-box (MAFbx) in skeletal muscle tissues [2]. 

MuRF-1 and MAFbx are key regulators of muscle atrophy, with MuRF-1 acting as an E3 ubiquitin ligase that targets specific proteins for degradation via the proteasome system. Its expression is upregulated in response to muscle damage, inflammation, and aging, making it a widely used marker of muscle atrophy [3]. Similarly, MAFbx, also an E3 ubiquitin ligase, promotes the degradation of muscle-related proteins, contributing to muscle wasting [4]. Both MuRF-1 and MAFbx are essential in regulating muscle mass by controlling the breakdown of muscle proteins under stress conditions [5]. Therefore, MuRF-1 and MAFbx are frequently used as biomarkers to assess myopathy.

Inflammatory myopathy refers to a condition where muscle mass is reduced due to an inflammatory response, particularly when an excessive immune system reaction, characteristic of autoimmune diseases, causes damage to muscle tissues and surrounding structures, leading to inflammation and subsequent muscle loss [6]. These muscle losses are strongly associated with autoimmune diseases and excessive inflammatory responses, which in turn can trigger a cascade of events disrupting muscle homeostasis [7,8]. Inflammatory mediators, such as tumor necrosis factor-alpha (TNF-α), interleukins, and prostaglandins, are often involved in amplifying the inflammatory response, causing direct damage to muscle fibers and promoting the activation of muscle-degrading pathways including MuRF1 and atrogin-1/MAFbx pathways [9]. Furthermore, inflammatory cytokines like TNF-α, interleukin-6 (IL-6), and interleukin-1β (IL-1β) can activate muscle-specific protein catabolic pathways such as the atrogin-1/MAFbx and MuRF1 pathways, which lead to accelerated protein degradation in skeletal muscle cells [10]. The onset of excessive inflammation is closely linked to muscle protein breakdown, muscle loss, and damage. To alleviate inflammatory muscle loss and its symptoms, high levels of steroids or immunosuppressant treatments were used. However, long-term steroid medication can lead to undesirable side effects such as osteoporosis, gastric ulcers, weight gain, and worsening of diabetes, and to reduce these side effects, immunosuppressive therapy is used in combination with other drugs to treat the inflammatory myopathy [11]. Due to the side effects, there has been an increasing effort to develop non-toxic and high functional ingredients from terrestrial and marine bioresources. In particular, ongoing research focuses on evaluating the protective effects of natural products derived from food or other biological sources against muscle loss, as well as exploring in vitro strategies to prevent or mitigate muscle atrophy [12,13]. More recently, various bioactive properties of natural compounds derived from seaweed have garnered increasing interest. Specifically, bioactive polyphenols derived from the extracts of *Ishige okamurae* (*I. okamurae*), including diphlorethohydroxycarmalol (DPHC) and polyphenolic compounds, have shown various bioactivities such as anti-obesity, anti-diabetic, vasodilatory, and UV-protective activities [14,15,16,17]. Recent studies have highlighted the potential of natural bioactive compounds, such as DPHC, derived from the marine brown alga *I. okamurae*, as a promising candidate for protecting against muscle loss and inflammation [4,18]. Our previous report demonstrated that DPHC exhibits strong protective effects against oxidative stress-induced muscle loss by downregulating MuRF-1/MAFbx expression, reducing muscle degradation, and suppressing nuclear factor kappa light chain enhancer of activated B cells (NF-κB) and mitogen activated protein kinase (MAPK)-associated inflammatory signaling pathways [10,18]. Building on these findings, further investigation was necessary to elucidate whether DPHC could exert similar protective effects against TNF-α-induced muscle loss particularly in a physiologically relevant in vivo model. The present study aimed to explore the protective potential of DPHC, derived from *I. okamurae*, against TNF-α-induced inflammatory muscle loss in an in vivo zebrafish model. Expanding on earlier research, the study provides further insight into the therapeutic potential of DPHC for inflammatory muscle loss and myopathy.

## 2. Results

### 2.1. Effect of DPHC on Swimming Performance of TNF-α-Induced Zebrafish

To investigate the potential effect of DPHC on swimming performance, we utilized a swim track system to assess swimming parameters such as swimming distance, upstream swimming time, and frequency of moving into Zone A in TNF-α-induced zebrafish. The top view of the swim track is shown in Figure 1A, and the side view of the swim track is shown in Figure 1B.

The results for the swimming parameters are presented in Figure 2. As shown in Figure 2A, the survival rate of all experimental groups demonstrated that there were no toxic effects on the zebrafish adults. Additionally, body weight did not show significant differences between the groups (Figure 2B). The potential swimming performance was assessed by measuring the number of zebrafish swimming, the distance swum, the time spent swimming, and the frequency of swimming in Zone A. Figure 2C demonstrates that TNF-α administration significantly decreased the number of zebrafish swimming in Zone A compared with the Control group. Treatment with DPHC effectively increased the number of zebrafish swimming in the upstream swim track compared with the TNF-α-treated group. Furthermore, OCT treatment (25 μg/g) also significantly increased the number of zebrafish swimming in Zone A compared with the TNF-α group, serving as a positive control. In addition, whether the DPHC treatment could improve swimming performance was further analyzed by measuring the distance swum, the time spent swimming in Zone A. Figure 2D reveals that the duration of time spent in Zone A did not differ significantly from Control. Similarly, no significant differences were observed in the DPHC-treated groups or the OCT-treated group. However, results regarding the distance moved (Figure 2E) show that the swimming distance significantly decreased in TNF-α-induced zebrafish adults. In contrast, both DPHC and OCT treatments significantly increased the swimming distance compared with the TNF-α-treated group. Moreover, the frequency of swimming in the swim track (Figure 2F) showed no significant difference due to TNF-α treatment. However, OCT treatment increased swimming frequency, confirming its role as a positive control for enhancing swimming performance.

### 2.2. Effect of DPHC on Inflammatory Signaling in Skeletal Muscle Tissue of TNF-α-Induced Zebrafish

To examine the protective effect of DPHC against TNF-α-induced inflammation, we analyzed the expressions of NF-κB and p38 MAPK in muscle tissue of TNF-α-induced zebrafish. As shown in Figure 3A, the Western blot bands of key NF-κB and MAPK-related proteins and the relative protein expressions were quantified. In Figure 3B–E, exposure to TNF-α significantly increased the protein expressions of phosphorylated p-IκB-α, p-NF-κB, p-JNK, and p-p38 compared to the Control group. However, DPHC treatment significantly decreased the expressions of p-IκB-α, p-JNK, and p-p38 in response to TNF-α exposure. OCT, as a positive control, reduced p-IκB-α, p-JNK, and p-p38 levels, indicating its potential to alleviate inflammatory responses.

### 2.3. Protective Effect of DPHC Against Inflammatory Muscle Loss: Modulation of MuRF-1 and MAFbx in Skeletal Muscle Tissue of TNF-α-Induced Zebrafish

To evaluate whether DPHC exerts a protective effect against TNF-α-induced inflammatory muscle loss, we analyzed muscle-degradation-associated proteins such as MuRF-1 and MAFbx in the skeletal muscle tissue of TNF-α-induced adult zebrafish. As shown in Figure 4A, the Western blot reveals protein bands for MuRF-1 and MAFbx. Figure 4B,C show a significant increase in the expression levels of MuRF-1 and MAFbx proteins in skeletal muscle tissue following TNF-α exposure in TNF-α-induced adult zebrafish. While the protein expression of MuRF-1 and MAFbx in the skeletal muscle tissue of TNF-α-induced adult zebrafish was elevated, it was effectively reduced by DPHC treatment. Similarly, OCT, used as a positive control, also significantly reduced the protein expressions of MuRF-1 and MAFbx, supporting its potential role in mitigating inflammatory muscle loss.

### 2.4. Protective Effect of DPHC on Skeletal Muscle Tissue Damage: Histological Analysis of TNF-α-Induced Inflammatory Muscle Loss

To further explore the protective effects of DPHC on skeletal muscle damage in TNF-α-induced zebrafish, histological analysis was performed using hematoxylin and eosin (H&E) staining. Figure 5 shows well-organized skeletal muscle fibers without structural abnormalities in the control group. However, the TNF-α-treated group showed significant muscle fiber degeneration, irregular arrangements, and structural damage, indicating severe inflammatory muscle loss. Meanwhile, DPHC treatment, at both low (2 μg/g) and high (5 μg/g) concentrations, improved the structural integrity of skeletal muscle tissue from TNF-α-induced zebrafish. Notably, the high-dose DPHC-treated group exhibited a remarkable restoration of muscle fiber alignment and organization comparable to the control group. Additionally, treatment with OCT (25 μg/g), a positive control, also mitigated muscle tissue damage, as indicated by the recovery of muscle fiber structure. These findings demonstrate the protective effect of DPHC on skeletal muscle under TNF-α-induced inflammation.

## 3. Discussion

Inflammation is a complex physiological response triggered by infections, tissue damage, or inflammatory mediators. Key pathways such as COX-2, NF-κB, and MAPK regulate pro-inflammatory cytokine production during inflammation [19]. Excessive inflammation causes cellular damage and tissue dysfunction [8]. In inflammatory muscle loss, excessive cytokines upregulate MuRF-1 and MAFbx, key regulators of muscle protein degradation [19]. MuRF-1 and MAFbx upregulation promotes muscle protein degradation, highlighting the need to target both inflammation and protein degradation to treat inflammatory muscle loss. In the present study, the protective effect of DPHC on TNF-α -induced inflammatory myopathy was investigated in an in vivo zebrafish model. The major research finding demonstrated that DPHC showed a significant protective effect against TNF-α-induced inflammation and muscle loss in skeletal muscle cells by regulating the NF-κB, MAPK, MuRF-1, and MAFbx signaling pathways. These results suggest that DPHC provides protective effects against inflammatory muscle loss by regulating NF-ĸB and MAPK associated signaling in muscle tissue of TNF-α-induced zebrafish, thereby reducing pro-inflammatory cytokines and inflammation. Consequently, the reduction of pro-inflammatory cytokines also led to the downregulation of MuRF-1 and MAFbx signaling pathways, contributing to the suppression of muscle loss. These findings are strongly supported by previous studies demonstrating the anti-inflammatory potential of DPHC isolated from *I. okamurae* [20]. Earlier reports highlighted DPHC’s ability to effectively inhibit TNF-α activity by binding to it with high affinity and down-regulating key pro-inflammatory cytokines such as IL-6 and IL-1β [18]. These actions are mediated through the suppression of the NF-κB and MAPK signaling pathways, which play critical roles in regulating inflammation and inflammatory muscle loss.

Zebrafish (Danio rerio) are a cost-effective in vivo model suitable for large-scale experiments and high-throughput screening [21]. Zebrafish enable muscle performance assessment through behavioral and muscle-specific assays, such as swimming tests, to evaluate functional deficits from genetic mutations or drug treatments [22]. In the current study, we assessed the potential protective effects of DPHC against TNF-α-induced inflammatory muscle loss in zebrafish by measuring swimming performance in Zone A. The major research finding demonstrated that TNF-α treatment significantly decreased the numbers of swimming zebrafish in Zone A. In addition, the moved distance also significantly increased with low and high concentrations of DPHC administration similar with the OCT-treated group in Zone A. These results are similar to our previous report; the high concentration of DPHC treatment effectively improved swimming performance under H_2_O_2_-induced oxidative stress [4]. Based on our results, we suggest that DPHC effectively improved swimming performance and alleviated TNF-α-induced inflammatory muscle loss in zebrafish, highlighting its potential as a functional food or health supplement for improving inflammatory muscle loss.

## 4. Materials and Methods

### 4.1. Chemicals

All chemicals and reagents utilized in the study were of analytical grade: 4% paraformaldehyde, ethyl 3-aminobenzoate methanesulfonate (MS-222) and eosin Y solution (Sigma-Aldrich, St Louis, MO, USA), hematoxylin (DAKO, Glostrup, Denmark). The primary antibodies including p-IĸB-α, p-NF-ĸB, p-JNK, p-p38 were obtained from Cell Signaling Technology (Danvers, MA, USA). Antibodies for MuRF-1, MAFbx, and β-actin, along with secondary antibodies, were sourced from Santa Cruz Biotechnology (Santa Cruz, CA, USA). The Lamin B polyclonal antibody (catalog number: 16730) was purchased from Invitrogen (Carlsbad, CA, USA).

### 4.2. Isolation of DPHC from I. okamurae

The isolation and purification of DPHC were conducted as previously described by Kim et al. (2020) with slight modifications [23]. In March 2019, *I. okamurae* were collected from the coastline of Jeju Island, Republic of Korea, washed, dried, and ground into a fine powder. The powdered material was extracted with methanol at 37 °C for 24 h under continuous stirring, and the extract was filtered and concentrated using rotary evaporation. The concentrate was re-dissolved in deionized water and subjected to stepwise solvent extraction with n-hexane, chloroform, and ethyl acetate. The ethyl acetate fraction, enriched with DPHC, was further purified using high-performance centrifugal partition chromatography under previously described conditions. The isolation process yielded DPHC at 0.39% with a purity of over 85%, as reported in previous studies [18,23].

### 4.3. Western Blot Analysis

Skeletal muscle tissues were lysed in a buffer containing 100 mM sodium fluoride, 10 µg/mL aprotinin, 10 µg/mL leupeptin, 10 mM sodium pyrophosphate tetrabasic, 2 mM sodium orthovanadate, 1% NP-40, 5 mM ethylenediaminetetraacetic acid, 1 mM phenylmethylsulfonyl fluoride, and 20 mM Tris for 1 h. Homogenization was performed using a bead beater (Taco™ Prep, GeneReach Biotechnology Corp., Taichung City, Taiwan) with 1 mm steel beads for five 40 s cycles, followed by centrifugation at 12,000 rpm for 15 min at 4 °C. Protein concentrations were determined using the Pierce™ BCA Protein Assay Kit (Thermo Fisher Scientific, Waltham, MA, USA) with bovine serum albumin (BSA) as a standard, and sample protein concentrations were adjusted to 20–30 µg/mL. Equal amounts of protein (15 µL) were separated by SDS-PAGE and transferred onto nitrocellulose membranes. Membranes were blocked with 0.5% skim milk in 1X Tris buffered saline with the addition of tween 20 (TBST) for 2–3 h, incubated overnight at 4 °C with primary antibodies (1:1000, Cell Signaling Technology), and then incubated with secondary antibodies (1:3000, Santa Cruz Biotechnology) for 1–2 h. After TBST washes, protein bands were visualized using enhanced chemiluminescence and imaged with a Fusion Solo system (Vilber Lourmat, Marne-la-Vallée, France).

### 4.4. In Vivo Design

Adult zebrafish were procured from a commercial supplier at the World Fish Aquarium, Jeju, Republic of Korea, and were bred following the procedures described by Lee et al. (2024) [4]. All animal handling and experimental procedures were conducted in accordance with the ethical guidelines approved by the Institutional Animal Care and Use Committee of Jeju National University (Approval No. 2022-0005, 17 April 2023). A total of 75 adult male zebrafish (15 per group) were randomly assigned to the following groups: control group (normal), TNF-α-exposed group (5 μM), DPHC-treated groups (2 and 5 μg/g, co-treated with TNF-α), and octacosanol (OCT)-treated group (25 μg/g, co-treated with TNF-α). The zebrafish oral administration procedure was performed according to the previously reported method [24]. For the oral administration, the zebrafish were anesthetized with 0.03% MS-222. Then, the anesthetized zebrafish were put into 1.5 mL Eppendorf tubes containing DPHC for 2 min. After 2 min, the DPHC-treated zebrafish were released into the tank. The DPHC treatment was performed three times a week for 2 weeks. At the end of week 2, DPHC and TNF-α were also administered orally using the same method to induce myopathy. At the end of week 2, following the third administration of DPHC and TNF-α and the swimming test, all zebrafish were sacrificed to collect skeletal muscle tissue for further tissue analysis.

### 4.5. Survival Rate, Body Weight, and Swimming Performance Assessment in Zebrafish

To assess the effects of DPHC on zebrafish, survival rates, body weight and swimming performance were measured in an in vivo zebrafish study. Survival rates were determined by counting the number of live zebrafish adults. For body weight measurements, zebrafish were first anesthetized using a 0.03% MS-222 solution. The zebrafish were then placed in a swimming tank with a 30 L water capacity, featuring adjustable flow and slope. This setup encouraged the fish to swim actively to maintain their position. The tank was divided into two zones: Zone A, designated as the upstream swimming track, exposing the fish to the water flow and gradient, and Zone B, which acted as the resting zone. The water temperature was maintained at 28.5 °C to match the acclimation temperature. Swimming performance was evaluated by counting the number of zebrafish remaining in Zone A after a set duration. Additionally, the time spent and distance covered by the zebrafish in Zone A were recorded using top-view video footage captured with the Loligo System (Viborg, Denmark).

To evaluate the effects of DPHC on zebrafish, survival rates and body weights were measured during in vivo experiments. Zebrafish were anesthetized with 0.03% MS-222 and placed in a 30 L swimming tank with adjustable water flow and slope to induce active swimming. The tank was divided into an upstream swimming track (Zone A) and a resting zone (Zone B), with water temperature maintained at 28.5 °C. Swimming performance was evaluated based on the number of zebrafish swimming, endurance time, swimming distance, and swimming frequency in Zone A, which were quantified using top-view video recordings obtained with the Loligo System (Viborg, Denmark).

### 4.6. Hematoxylin and Eosin Staining

To assess the oblique section of muscle tissue, which allows for the simultaneous evaluation of both the orientation and density of muscle fibers, H&E staining was carried out on paraffin-embedded muscle tissue sections. The sections were first deparaffinized and rehydrated, followed by staining with hematoxylin solution (DAKO, Glostrup, Denmark) for 30 s to 1 min to highlight nuclei. After rinsing with distilled water for 10 minutes, the slides were treated with eosin Y solution (Sigma-Aldrich, St. Louis, MO, USA) for 30 s to 1 min at room temperature to stain the cytoplasm. Nuclei were identified by their blue color, while the cytoplasm appeared light pink. The stained tissue slides were observed using a light microscope (Olympus Optical Co., Tokyo, Japan), and the CSA of the muscle fibers was measured using Image J (1.53 k) software (National Institutes of Health, Bethesda, MD, USA). Histological evaluations were performed in a blinded manner, with at least three independent analysts conducting triplicate assessments for each sample.

### 4.7. Statistical Analyses

All experiments were conducted in triplicate, with 15 zebrafish per group (*n* = 15), and data analysis was performed using one-way analysis of variance (ANOVA) with SPSS software (version 12.0). To evaluate significant differences among group means, Tukey’s test was employed as a post hoc analysis. Statistical significance thresholds were set at # *p* < 0.05, ## *p* < 0.01 as compared to the untreated (control) group. * *p* < 0.05, ** *p* < 0.01 as compared to the TNF-α-treated group.

## 5. Conclusions

In conclusion, DPHC improved swimming performance (number of swimming zebrafish and swimming distance in Zone A) in TNF-α-induced zebrafish, primarily by suppressing NF-κB and MAPK associated proteins and muscle-degrading proteins (MuRF-1 and MAFbx) in muscle tissues. These findings highlight the potential of DPHC as a therapeutic candidate for treating muscle atrophy and inflammatory myopathies by offering a safer and more effective approach to managing muscle loss and inflammation.

## Figures and Tables

**Figure 1 marinedrugs-23-00376-f001:**
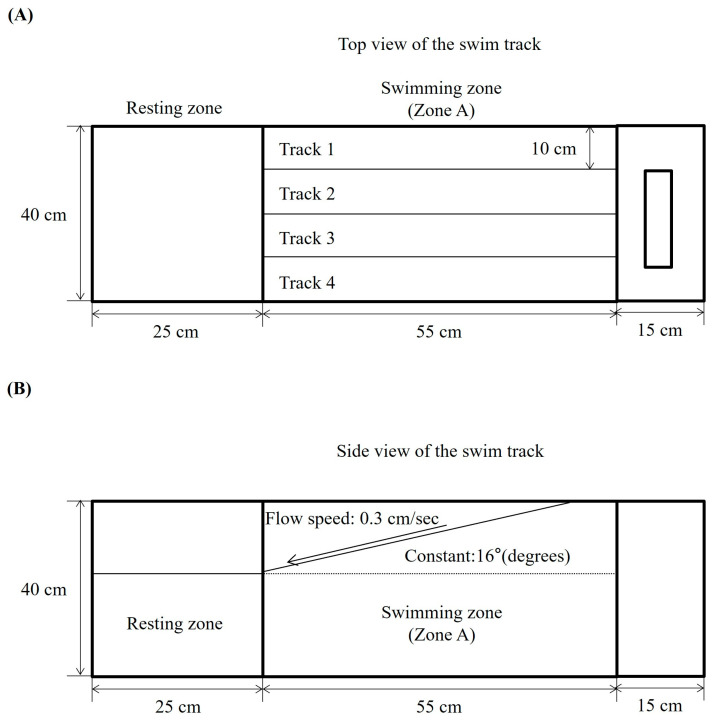
The schematic diagram of the zebrafish swim track for an endurance test against water flow and gradient. The behavioral endurance test was performed in the swim track, which was divided into two zones: Zone A, designated as the upstream swimming track, exposing the fish to the water flow and gradient, and Zone B, which acted as the resting zone. Constant water flow (0.3 cm/s) at an angle of 16° was maintained. The figures represent the swim track: (**A**) the top view and (**B**) the side view.

**Figure 2 marinedrugs-23-00376-f002:**
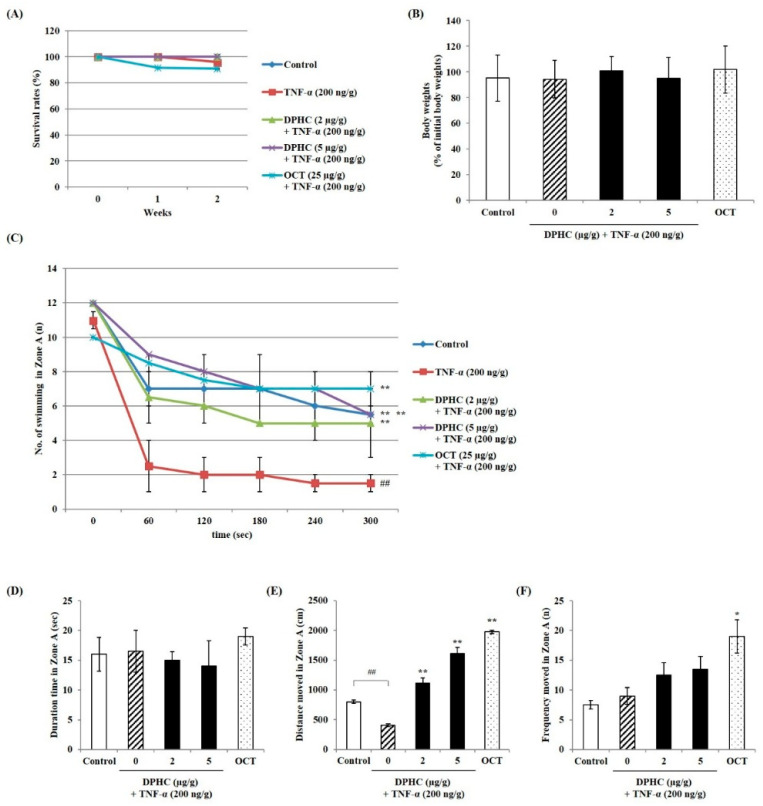
Effects of DPHC on behavioral endurance in TNF-α-induced zebrafish. (**A**) Survival rates, (**B**) body weights, (**C**) number of swimming zebrafish in Zone A, (**D**) duration time, (**E**) distance, and (**F**) frequency pattern in Zone A. Experiments were performed in duplicate with *n* = 15 zebrafish per group, and the data are expressed as mean ± SE; ## *p* < 0.01 as compared to the untreated (control) group. * *p* < 0.05, ** *p* < 0.01 as compared to the TNF-α–treated group.

**Figure 3 marinedrugs-23-00376-f003:**
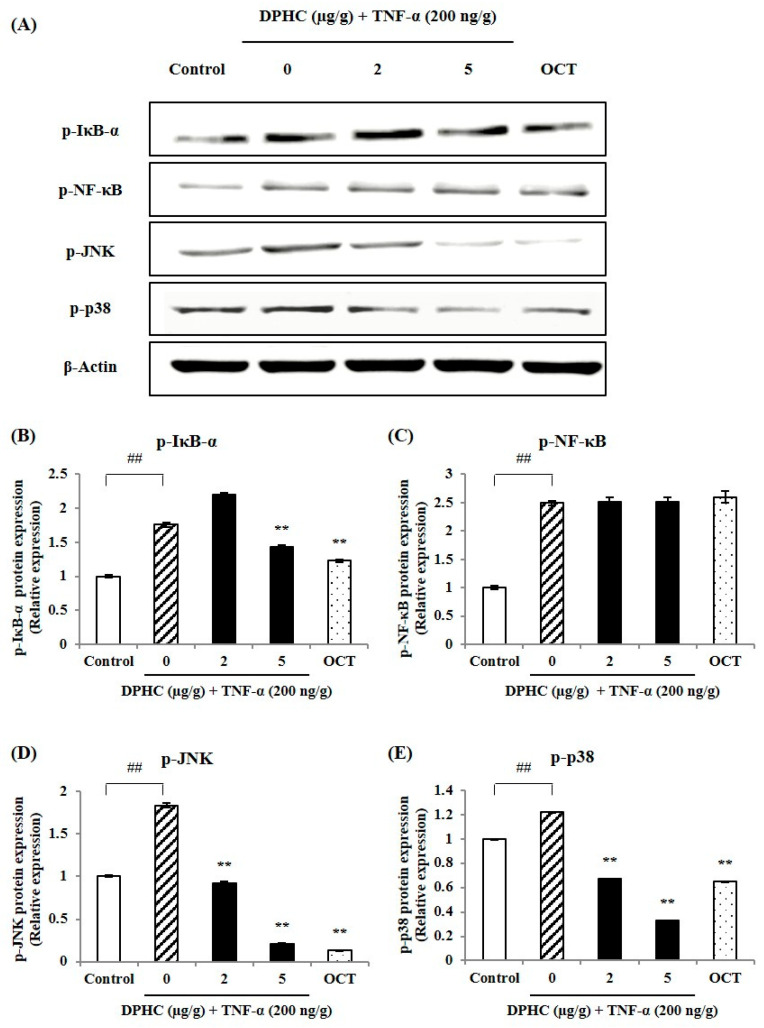
Effects of DPHC on NF-κB and MAPK associated proteins in skeletal muscle tissue from TNF-α-induced zebrafish. (**A**) Western blot bands of NF-ĸB and MAPK associated proteins and quantified (**B**) p-IĸB-α, (**C**) p-NF-κB, (**D**) p-JNK, (**E**) p-p38 expression. Experiments were performed in duplicate with *n* = 15 zebrafish per group, and the data are expressed as mean ± SE; ## *p* < 0.01 as compared to the untreated (control) group and ** *p* < 0.01 as compared to the TNF-α–treated group.

**Figure 4 marinedrugs-23-00376-f004:**
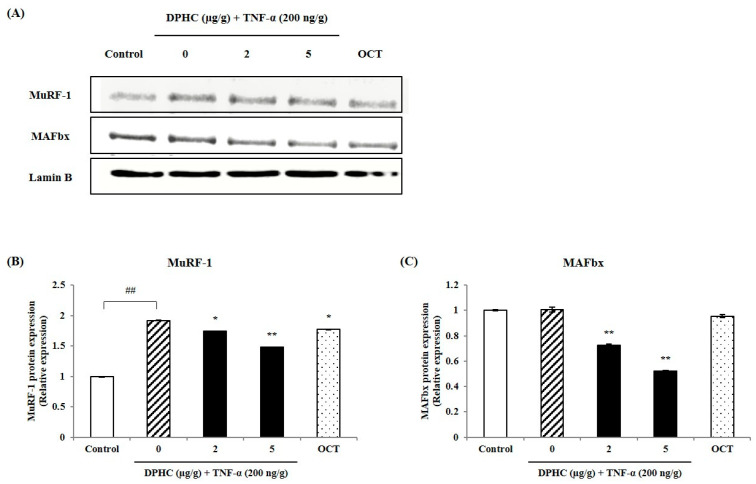
Effects of DPHC on muscle-degrading-related protein expression in skeletal muscle tissue from TNF-α-induced zebrafish. (**A**) Western blot bands of myopathy-associated proteins and quantified (**B**) MuRF-1, and (**C**) MAFbx expression. Experiments were performed in triplicate with *n* = 15 zebrafish per group, and the data are expressed as mean ± SE; ## *p* < 0.01 as compared to the untreated (control) group. * *p* < 0.05 and ** *p* < 0.01 as compared to the TNF-α–treated group.

**Figure 5 marinedrugs-23-00376-f005:**
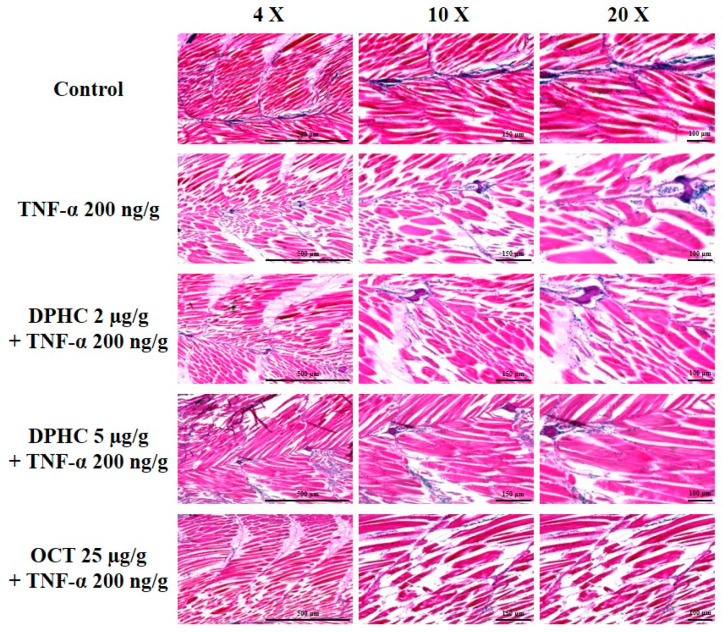
Effects of DPHC on skeletal muscle tissue loss in TNF-α-induced zebrafish. Skeletal muscle tissues were analyzed using hematoxylin and eosin (H&E) staining, where hematoxylin stains cell nuclei blue and eosin stains the extracellular matrix and cytoplasm pink. Experiments were performed with *n* = 15 zebrafish per group. Scale bars: 500 μm (4×), 150 μm (10×), 100 μm (20×).

## Data Availability

The data are available on request from the corresponding authors.

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
