# Peer review of "Therapeutic Potential of DPHC, A Brown Seaweed Polyphenol, Against TNF-α-Induced Inflammatory Muscle Loss"

_marinedrugs, 2025, doi:10.3390/md23100376_

Round 1
Reviewer 1 Report
Comments and Suggestions for Authors
1.Identify the first occurrence of "DPHC" in the manuscript (in the Title and Abstract) and define it as "diphlorethohydroxycarmalol (DPHC)". Ensure this full-name + abbreviation format is consistently applied for all subsequent first uses in major sections (e.g., Introduction, Materials and Methods) to align with standard biomedical publishing practices.
2.The manuscript contains two distinct sections labeled "Figure 4". Section 2.3 references "Figure 4" for Western blot data of MuRF-1 and MAFbx, while Section 2.4 later references another "Figure 4" for H&E staining results of skeletal muscle tissue-creating confusion about which figure corresponds to which data. A full check of all figures confirms inconsistent numbering and poor text-figure alignment.
3.Section 4.4 states that a total of 12 adult male zebrafish were allocated to 5 experimental groups , resulting in only 3 zebrafish per group. This sample size is far below the standard required for in vivo zebrafish studies (typically ≥6–8 individuals per group) and severely limits the statistical power of analyses for swimming performance, Western blots, and histological staining.
4.Section 4.2 describes the isolation of DPHC but omits essential details for reproducibility and quality control, such as the extraction yield of DPHC, the purity of the final DPHC sample, structural characterization data.
Author Response
Ms. Ref. No.: marinedrugs-3884089 B
Section / Special issue: SI: High-Value Algae Products, 2nd Edition
Title: Therapeutic potential of DPHC, a brown seaweed polyphenol, against TNF-α induced inflammatory muscle loss
Thank you very much for spending your valuable time in assessing our manuscript. We appreciate you detailed review and salient comments. We have carried out necessary modifications to the manuscript based on your comments. The changes are marked in red color within the manuscript.
Reviewer #1
1. Comment: Identify the first occurrence of "DPHC" in the manuscript (in the Title and Abstract) and define it as "diphlorethohydroxycarmalol (DPHC)". Ensure this full-name + abbreviation format is consistently applied for all subsequent first uses in major sections (e.g., Introduction, Materials and Methods) to align with standard biomedical publishing practices.
Response: Thank you for your valuable comment. We have defined diphlorethohydroxycarmalol (DPHC) at its first occurrence in the Abstract (Line 18–19) and in the Introduction (Line 74). The abbreviation has been applied consistently throughout the manuscript according to standard biomedical publishing practices. In addition, we have also defined nuclear factor kappa light chain enhancer of activated B cells (NF-κB), mitogen activated protein kinase (MAPK), interleukin-6 (IL-6) and interleukin-1β (IL-1β) by providing their full names followed by abbreviations at their first occurrences.
2. Comment: The manuscript contains two distinct sections labeled "Figure 4". Section 2.3 references "Figure 4" for Western blot data of MuRF-1 and MAFbx, while Section 2.4 later references another "Figure 4" for H&E staining results of skeletal muscle tissue-creating confusion about which figure corresponds to which data. A full check of all figures confirms inconsistent numbering and poor text-figure alignment.
Response: Thank you for pointing out this mistake. We have revised the content in Sections 2.3–2.4 and corrected the figure numbering and alignment to ensure consistency between the text and figures.
3. Comment: Section 4.4 states that a total of 12 adult male zebrafish were allocated to 5 experimental groups, resulting in only 3 zebrafish per group. This sample size is far below the standard required for in vivo zebrafish studies (typically ≥6–8 individuals per group) and severely limits the statistical power of analyses for swimming performance, Western blots, and histological staining.
Response: Thank you for your comment. We actually used 15 zebrafish per group, resulting in a total of 75 zebrafish across the 5 experimental groups. Accordingly, we have revised Sections 4.4 (Line 252) and 4.7 (Line 292-293) to clarify this point.
4. Comment: Section 4.2 describes the isolation of DPHC but omits essential details for reproducibility and quality control, such as the extraction yield of DPHC, the purity of the final DPHC sample, structural characterization data.
Response: Thank you for your valuable comment. The purity of the isolated and purified DPHC has been confirmed in our previous study, and we will provide the relevant reference below. In addition, we have revised the manuscript to include this information and cited the reference accordingly so that the purity and structural characterization of DPHC can be clearly verified. (Line 233-234)
Ref. High-performance centrifugal partition chromatography (HPCPC) for efficient isolation of diphlorethohydroxycarmalol (DPHC) and screening of its antioxidant activity in a zebrafish model, 2020, Process Biochemistry
Ref. Diphlorethohydroxycarmalol (DPHC) Isolated from the Brown Alga Ishige okamurae Acts on Inflammatory Myopathy as an Inhibitory Agent of TNF-α, 2020, Marine drugs
Reviewer 2 Report
Comments and Suggestions for Authors
In this study, the authors investigated the muscle loss reduction effect of polyphenols obtained from brown alga. This reviewer considers the following improvements essential for the publication of this paper.
- Has the zebrafish swimming assay presented in the manuscript been used in previously published studies? This manuscript did not provide sufficient information to determine whether this method is appropriate for evaluating swimming performance.
- The reason for using OCT as a positive control was unclear.
- Figure 2 : How many fish were used in swiming test? The legend for Fig. states duplicate. Meanwhile, L311 states triplicate (n=15). There is no consistency throughout the paper, making it unclear how many fish were used in the experiment.
- Figure 2: The method for administering the sample and TNFα to the fish cannot be reproduced based on the text provided in 4.4. In vivo design. The procedure must be described in a manner that allows the experiment to be replicated.
- Figure 3: It is unclear what time course was used for sample collection. For example, phosphorylation of NF-κB or MAPK is expected to vary significantly depending on the timing of sample collection. However, the experimental methods described in this study did not specify the drug exposure method or collection time for the samples.
- Figure 3: How many fish and extracted protein were used in these experiments?Please provide a clear explanation.
- L153-166 and L175-188 seems same sentence.
- lacks an explanation for Fig. 4
Author Response
Ms. Ref. No.: marinedrugs-3884089 B
Section / Special issue: SI: High-Value Algae Products, 2nd Edition
Title: Therapeutic potential of DPHC, a brown seaweed polyphenol, against TNF-α induced inflammatory muscle loss
Thank you very much for spending your valuable time in assessing our manuscript. We appreciate you detailed review and salient comments. We have carried out necessary modifications to the manuscript based on your comments. The changes are marked in red color within the manuscript.
Reviewer #2
1. Comment: Has the zebrafish swimming assay presented in the manuscript been used in previously published studies? This manuscript did not provide sufficient information to determine whether this method is appropriate for evaluating swimming performance.
Response: Thank you for your comment. We have added clarification in the Discussion (Lines 203–205) and cited relevant references demonstrating the application of zebrafish in studies of muscle-related diseases. Based on these references, we specified that the swimming assay can be appropriately used to evaluate muscle-related performance. The cited studies reported the use of zebrafish swimming patterns and behaviors to assess muscle function, and therefore we believe that the selected references are appropriate to support the validity of this method.
2. Comment: The reason for using OCT as a positive control was unclear.
Response: Thank you for your comment. We used OCT as a positive control in this study, and we provide the supporting reference(s) below to justify its use. This study was conducted in 2017. At that time, the Korean Ministry of Food and Drug Safety (MFDS) had designated octacosanol as an individually approved functional ingredient for muscle strength improvement. Therefore, OCT was selected as a positive control to validate the effectiveness of our experimental model. Therefore, OCT, approved by the MFDS, was used as a positive control to compare its effects on muscle strength with those of DPHC from the Brown Alga Ishige okamurae.
In addition, several studies have reported that octacosanol supplementation enhances exercise performance, endurance capacity, and muscle strength, supporting its appropriateness as a positive control in studies related to muscle function. The following reference(s) confirm the scientific rationale for using octacosanol as a positive control.
Ref. Diphlorethohydroxycarmalol (DPHC) Isolated from the Brown Alga Ishige okamurae Acts on Inflammatory Myopathy as an Inhibitory Agent of TNF-α, 2020, Marine drugs
Ref. Octacosanol Supplementation Increases Running Endurance Time and Improves Biochemical Parameters After Exhaustion in Trained Rats, 2004, Journal of Medicinal Food
Ref. Dietary Supplementation of Octacosanol Improves Exercise-Induced Fatigue and Its Molecular Mechanism, 2021, Food chemistry
3. Comment: Figure 2 : How many fish were used in swimming test? The legend for Fig. states duplicate. Meanwhile, L311 states triplicate (n=15). There is no consistency throughout the paper, making it unclear how many fish were used in the experiment.
Response: Thank you for your valuable comment, and we apologize for the confusion. In our study, 15 zebrafish were used per group for the swimming test. Therefore, the incorrectly stated parts have been corrected in the manuscript (Line 160, 253). In each experiment, 15 zebrafish per group (a total of 75 zebrafish) were used, and since three independent experiments were conducted, a total of 225 zebrafish were used.
4. Comment: Figure 2: The method for administering the sample and TNFα to the fish cannot be reproduced based on the text provided in 4.4. In vivo design. The procedure must be described in a manner that allows the experiment to be replicated.
Response: Thank you for your valuable comment. We have specified the method of administration for both the sample and TNF-α in the Materials and Methods section 4.4 In vivo design (Line 256–261).
Ref. Enzymatic Hydrolysates of Hippocampus abdominalis Regulates the Skeletal Muscle Growth in C2C12 Cells and Zebrafish Model, 2019, Journal of Aquatic Food Product Technology
5. Comment: Figure 3: It is unclear what time course was used for sample collection. For example, phosphorylation of NF-κB or MAPK is expected to vary significantly depending on the timing of sample collection. However, the experimental methods described in this study did not specify the drug exposure method or collection time for the samples.
Response: Thank you for your valuable comment. In this study, we evaluated whether DPHC could protect against TNF-α–induced muscle loss through an in vivo experiment. The experiment was conducted for a total of two weeks. During the first week, zebrafish received three administrations of DPHC, and in the second week, DPHC and TNF-α were co-administered orally three times. After the swimming test, skeletal muscle tissues were collected from zebrafish for further analysis. We have specified these details in the Materials and Methods section 4.4 In vivo design (Line 232–236).
6. Comment: Figure 3: How many fish and extracted protein were used in these experiments? Please provide a clear explanation.
Response: Thank you for your comment. We used 15 zebrafish per group in these experiments, and for the Western blot analysis, skeletal muscle tissue samples were prepared at a protein concentration of 20–30 µg/mL. Accordingly, we have specified the protein concentration in the Materials and Methods section 4.3 Western blot analysis (Line 240–242).
7. Comment: L153-166 and L175-188 seems same sentence.
Response: Thank you for your comment. We apologize for the confusion caused by our mistake. We have carefully re-checked the content and revised the sentences accordingly (Line 145–147).
8. Comment: lacks an explanation for Fig. 4
Response: Thank you for your valuable comment. This point has been addressed in our response to Comment 7.
Round 2
Reviewer 1 Report
Comments and Suggestions for Authors
The author has diligently addressed the reviewers' comments, suggesting that further refinement of the research mechanism should be pursued in subsequent studies.